# Assessing the relationship between digital technology use and physical health, fitness, and exercise levels among Chinese youth: The moderating effect of parental monitoring

Ziyi Hao[1]*, Jiajia Cui [2]*

**1** University of London Institution of Education, Gower St, London, United Kingdom, **2** Jilin Sport University, Changchun City, Jilin Province, China

\* 15643160000@qq.com (J.C.); fhxtt411@163.com (Z.H.)

## Abstract

The study investigates how digital media use, parental supervision, and attitudes towards physical activity influence young Chinese individuals' physical health and exercise levels. Concerns over the health impacts of increasing digital technology use among teenagers have spurred extensive research. However, the specific roles of parental supervision and personal attitudes towards physical activity in mitigating or exacerbating these effects remain unclear. The study sample comprises 827 Chinese youth from diverse geographical locations. Regression analyses highlight significant predictors of physical health and exercise levels, including digital technology use ($\beta = 0.35$, $p < 0.001$), social media use ($\beta = 0.22$, $p = 0.003$), online gaming ($\beta = 0.19$, $p = 0.011$), educational technology use ($\beta = 0.29$, $p = 0.003$), parental monitoring ($\beta = 0.47$, $p < 0.001$), and attitudes towards physical activity ($\beta = 0.50$, $p < 0.001$). Additionally, the result illustrates that attitudes towards physical activity moderate the relationship between digital technology usage and physical health and exercise levels (Indirect Effect $= 0.12$, $p = 0.003$). The moderation analysis shows that digital technology use, social media use, online gaming, and educational technology use exhibit significant interactions with parental monitoring, indicating that parental supervision can mitigate the health risks associated with digital device usage. These findings underscore the crucial role of parental involvement in mitigating the health risks associated with adolescent technology use.

## 1. Introduction

Digital technology has pervaded contemporary life, affecting people's behaviours and lifestyles, particularly young people. The internet, mobile devices, and tablets have transformed social, educational, and recreational activities. Social networking, online gaming, and instructional technology are examples of new entertainment brought

**Data availability statement:** All the data are made available in the repository database through https://doi.org/10.6084/m9.figshare.28061969.v1

**Funding:** The author(s) received no specific funding for this work.

**Competing interests:** The authors have declared that no competing interests exist.

about by the digital revolution, which has enhanced communication and information access[1]. However, young people frequently use digital gadgets, which may damage their fitness, health, and exercise habits. This circumstance requires studying the relationship between digital technology use and physical health and how parental supervision moderates it[2].Young people's lives have been transformed by digital technology. Cheap and accessible digital tools are causing teens and young people to spend more time online. Young people use WeChat, Weibo, and TikTok to stay in touch and keep up with current events. These platforms provide social engagement and self-expression but promote screen time and inactivity. Internet gaming has proliferated due to its interactivity and immersion[3]. Playing online video games may improve memory and socialize via multiplayer features, but too much might harm health.Digital technology has transformed education, changing how individuals learn. Digital tools are being used in schools to help students learn and succeed. Online learning platforms, educational apps, and digital materials make education more accessible and engaging than ever[4]. Digital technology for education may increase screen time, affecting students' physical activity. As they balance schooling, online socializing, and entertainment, young people must understand how digital technology impacts their physical health.As young people are in rapid physical and mental growth, they must prioritize their physical health, fitness, and exercise as part of a balanced lifestyle[5]. Regular exercise is critical to maintaining heart health, muscular strength, and overall well-being. It is a powerful tool in the fight against obesity, diabetes, and hypertension. However, the current generation of teens and young adults is less active, possibly due to their technology addiction. Given that sedentary behaviours have been linked to health problems, it is imperative to encourage young people to be more active and adopt a balanced lifestyle[6].

Parental supervision plays a crucial role in shaping children's technology use and health. Parents have a significant influence on their children's behaviour. By setting restrictions and standards for technology use, parents can actively contribute to their children's physical health. It is important for parents to monitor their children's screen time, encourage outdoor play, and emphasize the value of a balanced lifestyle[7]. Given the varying approaches to parental participation and monitoring, it is essential to study how parental supervision can effectively reduce the potentially harmful impacts of teenagers' digital technology use on their physical health.Teens' exercise attitudes and parental oversight moderate this link. Exercise and fitness beliefs substantially influence a person's physical activity[8]. Positive attitudes encourage more individuals to exercise and maintain an active lifestyle. Negative attitudes or a lack of motivation to exercise may lead to sedentary habits and poor health. It is time to understand how digital technology affects young people's exercise habits to help them live healthier[9].

The rapid growth of technology and the cultural emphasis on academic performance make China an excellent venue to explore these dynamics. Due to the rise of internet access and digital devices, Chinese youth utilize digital technology exponentially. Like others elsewhere, Chinese youth are extensively immersed in social media, gaming, and educational technologies[10]. Cultural emphasis on education

and high academic standards lead to lengthy study hours and increased screen use. This issue underscores the necessity of studying China's youth's digital technology use and physical health.Studies show Chinese parents worry about technology's health and safety implications on children[11,12]. In China, parents restrict screen time and promote healthy lifestyles in the classroom and beyond to monitor their children. However, parental health understanding, physical activity resources, and education may alter these practices' effectiveness. Studying the effects of parental supervision on digital technology use and physical health may help Chinese youth establish balanced lives[13].Cultural and social factors influence young Chinese people's views on exercise. The prevailing notion is that students should prioritize academics over-exercise. However, public health campaigns and government initiatives are helping individuals recognize how vital body care is. Understanding how digital technology affects young Chinese people's ideas about physical activity may help promote them[14].

There is considerable concern regarding the potential detrimental effects of adolescents' extensive use of digital devices on their physical and mental well-being. Outdoor activities and fitness were restricted during the COVID-19 pandemic, prompting individuals to migrate to digital media. Peng et al. [15] investigated the dual application of digital tools in health management by examining the manner in which virtual reality fitness (VRF) mediated the relationship between physical activity and well-being during the pandemic. Fang et al. [16] assessed demographic parameters that influence the use of digital platforms for physical activity, which can either promote or discourage active lifestyles. The manner in which Chinese individuals exercise has been altered by the proliferation of fitness applications and online workout programs [17]. Nevertheless, there is a lingering concern regarding physical inactivity and computer use, particularly among newer generations. Yang et al. [18] demonstrated the potential of VRF to regulate behavior and promote physical activity during the pandemic, while Saqib and Qin [19] underscored the significance of digital advancements in operational and physical health. Saqib et al. [20] and Dai and Men has [21] found that physical activity promotes the health of Sustainable Development Goals (SDGs) and mitigates chronic diseases.

This investigation concentrates on three primary research questions regarding the correlation between the amount of digital technology youths use and their physical health. First, to determine how various digital technologies affect Chinese youth fitness, health, and exercise habits. These include social networking, internet gaming, and instructional applications. Analyzing these relationships will illuminate how different types of youthful internet contact support or discourage active living. Second, examine how parental supervision may reduce technology's detrimental impacts on fitness. The study examines how parenting practices like screen time limits and physical exercise affect children's health. These dynamics must be understood to help parents reduce the negative effects of digital technology on their children's physical health. Third, to examine teens' digital media use and physical health via their exercise habits. This explores young people's viewpoints and motives for playing sports and exercising to uncover factors that promote good physical activity habits. These research questions aim to explain the complex link between young Chinese people's physical health, digital device usage, parental influence, and exercise habits. The study's research objectives are as follows:

1. To examine the association between digital technology use and physical fitness, health, and exercise levels among Chinese youth.

2. To assess the moderating impact of parental monitoring on the relationship between physical health and digital technology use.

3. To delve into the academic significance of youths' attitudes towards physical activity, investigating its mediating role in the relationship between digital technology use and physical health among Chinese youth.

The study's findings aid parents, educators, lawmakers, and healthcare professionals. Overseeing children's screen time diminishes health hazards, encourages improved habits, and enhances physical fitness; these findings can assist educators in formulating technology-integrated pedagogical strategies that improve physical activity and digital education.

Further, the findings assist policymakers in developing recommendations and initiatives to restrict adolescent digital technology usage, promote active lifestyles, and mitigate sedentary behaviors associated with screen time. Healthcare professionals can utilize the findings to develop targeted health promotion initiatives that highlight parental engagement and tackle issues related to digital media, physical activity, and overall health.

The study is divided into the following sections: The introductory part is located in section 1. Section 2 contains the literature review. The materials and procedures are presented in section 3. The findings are analyzed and explained in section 4. The last section proceeds with the examination.

## 2. Literature review

This study holds significant implications for the field, aiming to illuminate the intricate connections between young Chinese people's use of digital technologies and their levels of physical activity, fitness, and overall health. In today's highly digitalized society, understanding the effects of smartphone use, social media, online gaming, and educational technology on young people's physical health is paramount. This literature review aims to consolidate previous studies that have delved into the effects of digital technology use, parental supervision, and adolescent attitudes towards physical exercise on the physical health outcomes of young people in China.

### 2.1. Theoretical framework

The theoretical framework that guides the study explains the complex interactions between its variables, drawing on many fundamental principles from earlier research. This approach relies on the Social Cognitive Theory [22], which asserts that complex internal and external forces affect behaviours. Social interactions and cognitive processes shape behaviours; individuals learn by watching, mimicking, and being rewarded. Parental modelling and contextual factors affect youth digital technology and fitness habits. Social cognitive theory illuminates these impacts in this study. The parental monitoring hypothesis [23] helps the framework comprehend how parents influence their children. Good parents restrict their children's screen time and exercise in addition to watching them. Research shows that children of active parents are less likely to be sedentary and more willing to exercise [24,25]. This association is mediated by young physical activity attitudes, explained by the Theory of Planned Behavior[26]. This theory says attitudes, subjective norms, and behavioural control perceptions influence behavioural intentions and acts. An optimistic view on exercise may lessen the adverse effects of screen time on physical health [27].

Recent research reveals that digital technology is having an impact on people's physical health, particularly among young people. de Lima et al. [28] found a significant association between excessive digital device use and negative physical health indicators such as reduced physical activity and sedentary behavior. According to the Social Cognitive Theory (SCT), external factors, particularly digital technology, influence people's activities, including their own. According to Xie et al. [29], parental control is substantially related to children's activity levels. Banić&Orehovački[30]found that parental monitoring significantly reduces digital technology's impact on children's health behaviors. The Theory of Planned Behavior (TPB) can help explain physical activity decision-making, as Meng et al. [31]found attitudes toward technology use had a substantial impact on participants' physical activity levels. TPB includes attitudes, subjective norms, and perceived behavioral control. Cultural and familial expectations influence young Chinese people's internet involvement [32]. These findings have policy and practical implications because they highlight the need for culturally relevant treatments that address both technology use and physical health. The proposed theoretical framework links the variables, as indicated in Fig 1.

Fig 1 illustrates that when parents monitor their children, technology may not affect their physical health. Teenage digital device use and physical health outcomes in China are influenced by their positive exercise attitudes. Parental oversight and physical activity attitudes shield Chinese youngsters from the health risks of digital device use. Understanding how Chinese adolescents see and handle physical exercise is critical to understanding how these factors affect the relationship between adolescent digital media use and health consequences.

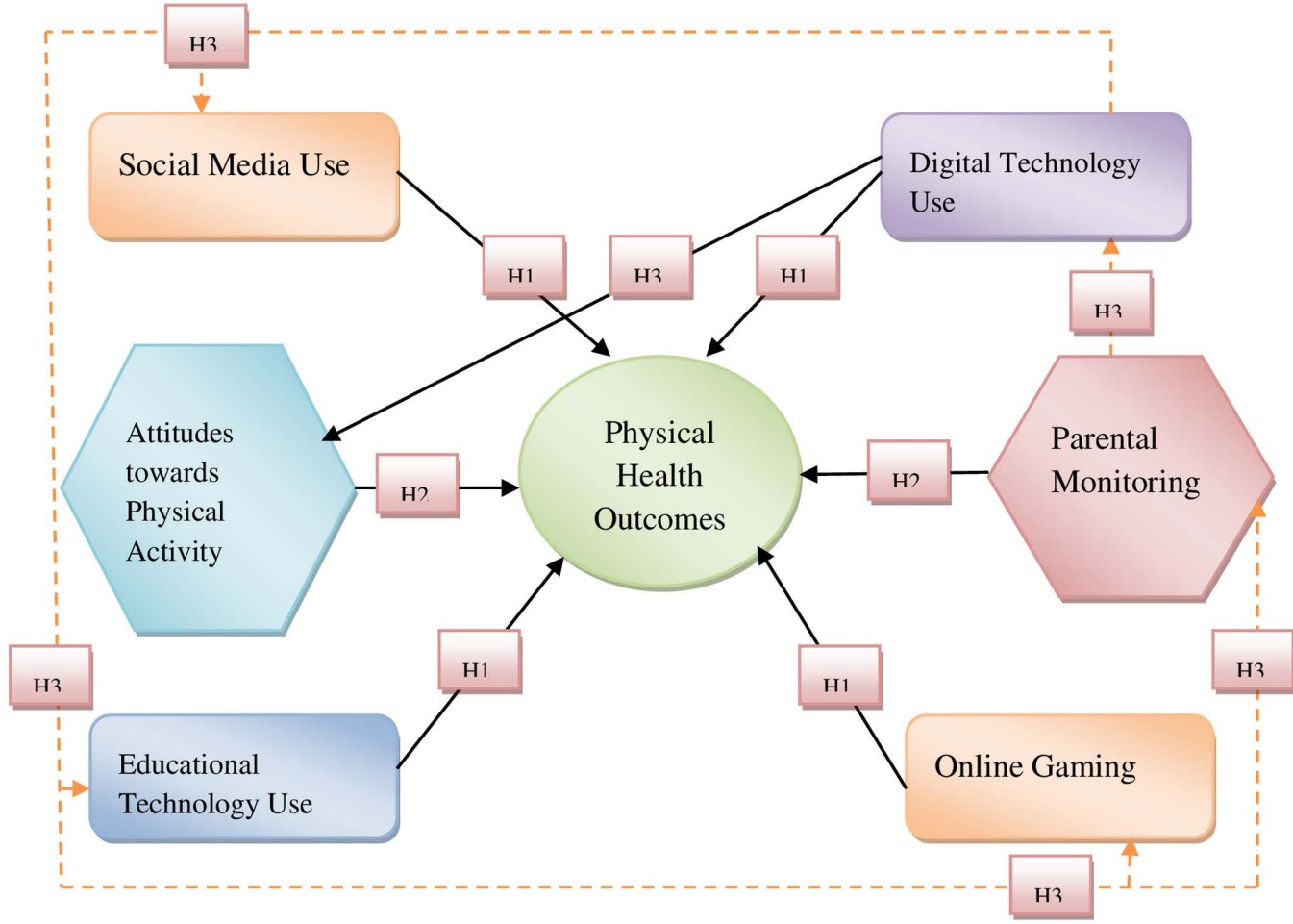

**Fig 1. Conceptual Framework.**

The fundamental point of the argument is that parental monitoring and physical activity attitudes moderate and mediate the relationship between digital device use and physical health. Digital technology affects sedentary behavior and physical health [33]. Kim et al. [34] and Fonvig et al. [35] found that social media, online gaming, and educational technology increase screen time, which reduces physical activity and increases health risks.

H1: The physical health and fitness levels correlate negatively with increased engagement in digital technology, encompassing general usage and social media platforms.

It is generally known that sedentary behaviours like screen time reduce fitness and health [36]. Research links social media use to increased screen time, which reduces physical activity [37].

H2: The significant use of educational technology and frequent online gaming negatively impact physical health and fitness levels.

Like online gaming, instructional tools can make people less active [38]. Academic and recreational screen use must be moderated for a healthy lifestyle [39].

H3: The correlation between physical health and digital technology utilization is influenced by favourable perceptions of physical activity, and proficient parental oversight moderates this relationship.

Despite rising internet use, a positive outlook can drive youth to exercise [40]. Research also shows that parents can assist youngsters in limiting their online usage and improving their habits [41]. These components affect health outcomes as environmental and behavioral buffers.

## 2.2. Impact of general digital technology and social media use on physical health

The widespread availability of smart phones and social media sites like WeChat and TikTok has profoundly impacted young people's habits and way of life. Multiple studies have shown that social media and overall digital technology usage negatively affect people's physical health[42,43]. Li et al. [44] stated that teenage screen usage is correlated with lower levels of physical activity, which adds to the epidemic of obesity and cardiovascular illness. In addition, social media addiction may cause extended inactivity and poor sleep quality, which worsens health problems [45]. Adolescents who spend too much time on social media are more likely to have adverse mental health outcomes and be less active as adults[46]. Digital technology promotes social connection, but Ni Shuilleabhain et al. [47] caution that it may also increase body dissatisfaction and social comparison, harming physical health. Goyal [48] found that digital technology makes kids sedentary. They suggest that screen time-reduction interventions improve physical health. Granet et al. [49] found social media reduce physical activity and health advantages as individuals exercise less. Dzhambov et al. [50] address social media's mental health effects. They note that stress and concern may indirectly affect physical health, discouraging kids from exercising more. Piores et al. [51] found that parental restrictions on social media and digital gadget use increase adolescent physical activity. Thorsteinsson et al. [52] states that digital media has caused teens to spend less time outdoors and with friends, damaging their physical health.

## 2.3. Effects of online gaming and educational technology use on physical health

The physical health of young people is affected in different ways by different aspects of digital involvement, such as instructional technology and online gaming. While playing video games online may be a social and cognitive boon, studies show that playing too much might cause people to become less active and more sedentary [53,54]. Ezechi[55] stated that gamers may be less likely to engage in real-life physical activities due to the immersive nature of gaming settings, which might lead to a sedentary lifestyle and related health hazards. On the other side, students now spend more time in front of screens, which might mean less time for physical exercise, thanks to the proliferation of digital tools in classrooms [56]. This contradiction highlights the need to research the effects of educational technology on young Chinese people's physical fitness and exercise habits, as well as their experiences with online gaming. Merino-Campos et al. [57] found that adolescents' usage of educational technology improves academic performance but decreases physical activity. Panjeti-Madan & Ranganathan [56] recommend parenting their children's screen time and physical activity by supervising and controlling their use of educational technology and online gaming. Tong et al. [58] investigate if educational technology might encourage physical exercise to minimize students' sedentary behaviours. Educational technology offers many new methods to learn. Liang et al. [59] encouraged children and adolescents to exercise using active gaming technology. Understanding gaming's social and psychological dimensions is crucial to encouraging younger gamers to embrace healthier lifestyles [60].

## 2.4. Role of parental monitoring and youths' attitudes as moderators and mediators

The correlation between teenagers' use of digital technologies and their physical health is heavily influenced by parental monitoring procedures and their attitudes towards physical exercise. Research has shown that when parents actively supervise their children's screen time and promote outside activities, it leads to better lifestyle choices and increased physical activity levels in teenagers [61]. Watson et al. [62] found that sedentary habits and health risks may worsen when parents do not adequately supervise their children's internet participation or have permissive views. Another factor influencing young people's exercise and sports participation is their attitude toward physical activity, which is shaped by societal standards and cultural views [63]. Higher exercise levels and better physical health outcomes are linked to positive

attitudes towards physical activity, whereas negative attitudes may inhibit involvement [64]. Lowe et al. [65] found that unsupervised teen digital device usage leads to sedentary behaviour and weight increase. Uddin & Hasan [66] found that teens whose parents restrict media and screen usage had better sleep and mental health. Parental physical activity and lifestyle modelling improve children's exercise habits and attitudes [67]. Vilardell-Dávila et al. [68] suggest that parental involvement and positive reinforcement increase youth physical activity and minimize sedentary behaviour.

### 2.5. Research gap(s) and contribution of the study

In China, the perspectives of youth on physical activity, their usage of digital technology, parental oversight, and their physical health and exercise levels pose significant challenges. A review of the literature suggests several areas requiring further investigation. Although prior research has shown that screen use adversely affects physical health, there is a paucity of longitudinal studies examining the enduring impacts of adolescent screen dependency[69,70]. Longitudinal research could shed light on whether findings linking screen use to poorer psychological well-being, as observed by Messena&Everri[71], persist into adulthood. Furthermore, there is limited research on how different types of digital technologies, such as online gaming and educational tools, impact the physical health of young Chinese individuals. While Alanko[72] discuss the positives and negatives of video gaming, including its effects on physical activity, further research is necessary to distinguish between recreational gaming and educational technology used in classrooms. The rise in educational technology use in classrooms may influence students' screen time and activity levels, underscoring the need for this distinction [73]. Additionally, exploring how parental supervision can mitigate the adverse effects of technology on children's physical health is crucial. Santos et al. [74] emphasize the role of parental support in promoting physical activity and reducing sedentary behaviour, yet little is known about how varying degrees and types of parental monitoring impact these outcomes. Comparative studies across different cultural contexts and age groups regarding screen time limits and strategies for promoting outdoor activities are needed. Finally, there needs to be more understanding how adolescents' attitudes towards physical activity mediate the relationship between their computer use and health outcomes. Newsome et al. [75] highlight the influence of attitudes and perceptions on adolescents' exercise routines, suggesting that positive attitudes towards physical activity may mitigate the sedentary effects of digital engagement. However, further investigation is needed to elucidate how attitudes towards physical activity mediate the link between adolescent digital device use and health outcomes in China.

The study addresses these gaps by examining the relationships between youths' attitudes towards physical activity, parental monitoring practices, dynamics of digital technology use, and Chinese youth's physical health and exercise levels. This research seeks to provide insights into these interconnections by employing a mixed-methods approach involving quantitative surveys and qualitative interviews. Given that existing research predominantly focuses on Western samples, this study's emphasis on Chinese youth provides valuable cultural context. Cultural factors such as parental expectations regarding physical exercise and societal norms surrounding digital technology use may significantly influence the health outcomes of Chinese youth. This study aims to inform the development of culturally appropriate internet usage guidelines for young Chinese individuals. Furthermore, the findings of this study may influence policies and interventions aimed at optimizing adolescents' use of digital technology for physical exercise and overall well-being. By enhancing parental monitoring practices and understanding attitudes towards physical activity, stakeholders in education, public health, and government sectors can devise tailored interventions to mitigate the potential adverse effects of digital technology on adolescent health.

## 3. Materials and methods

The survey included 12–18-year-old Chinese adolescents from urban and rural locations, the participants recruited between January to August 2023.To capture regional perspectives, the sample was drawn from Beijing, Shanghai, Guangdong, Sichuan, and Jiangsu. The findings should reflect China's diverse cultures and socioeconomic positions by

covering many locations. The Cochran sample size formula, specifically designed for studies involving large populations, is employed to ascertain the suitable sample size for this research. The equation is as follows:

$$n_0 = \frac{Z^2 \times p(1-p)}{e^2}$$

(1)

Where,
- Z represents the Z-score associated with the specified confidence level (e.g., 1.96 for 95% confidence).
- $p$ is the estimated proportion of the population exhibiting the trait of interest is generally established at 0.5 to optimize variability, such as devotion to physical activities, and
- $e$ represents the margin of error, generally set at 5% or 0.05.

Using these parameters, the study desired sample space is as follows:

$$n_0 = \frac{(1.96)^2 \times 0.5(1-0.5)}{(0.05)^2} = 384$$

The formula is used to alter the initial sample size ($n_0$) for a finite population is as follows:

$$n = \frac{n_0}{1 + \frac{n_0-1}{N}}$$

(2)

Here, N is represented by the population size. The sample size remains roughly 384, as the correction factor is minimal due to the large size of the target group (Chinese adolescents utilizing digital technologies).

The study's sample size of 827 participants exceeded this threshold, ensuring enhanced reliability and statistical power. This strategy aligns with the methodologies utilized by researchers such as Nam [76] and Kotrlik et al. [77], which employed analogous computations to determine dependable sample sizes.

The 1200 structured questionnaires distributed to Chinese youth had a 69% response rate, with 827 returning them. Respondents' demographics matched the population's gender distribution with negligible deviations. Early and late adolescents (12–14 and 15–18 years old) were categorized. The participants' educational background, household socioeconomic status, and urban or rural residence were also recorded. The sampling technique employed stratified random sampling, which ensured a representative sample from each demographic category, including age, gender, and urban/rural status. This method makes the study's findings more generalizable. A standardized questionnaire with variable-specific questions was utilized to collect study data. Questions like "I feel physically healthy and fit" and "I engage in regular physical exercise." assessed physical health, fitness, and activity. The Physical Activity Questionnaire for Adolescents (PAQ-A) was used to create these components to assure reliability and validity. The four independent variables were:

1. The digital technology use scale included "I spend a significant amount of time using digital devices (e.g., smartphone, tablet, computer)" and "My daily routine involves the use of digital devices." The Social Media Use Questionnaire (sMUQ), which inspired the items, was confirmed by Kircaburun et al. [78].

2. Social media use is assessed by stating, "I frequently use social media platforms (e.g., WeChat, Weibo, TikTok)" and "I find myself losing track of time when using social media." Jenkins-Guarnieri et al. [79] designed and validated the Social Media Use Integration Scale (SMUIS).

3. Online gaming measures consider answers to "I play online games regularly" and "I sometimes prioritize online gaming over other activities." The Online Gaming Questionnaire (OGQ), confirmed by Uçar [80], provided these questions.

4. Educational technology use wasassessed with questions such as "I use digital technology to assist with my schoolwork and studies" and "Educational apps and tools help me learn more effectively." The Educational Technology Use Survey (ETUS), proven by Pittman & Gaines [81], was used.

Parental monitoring was quantified by asking, "My parents set rules regarding my use of digital technology." and "My parents monitor the amount of time I spend on digital devices". Parental Monitoring Scale validation research, such as those by Stattin and Kerr [82], has led to this issue. Youth attitudes about physical activity—the mediator variable—were measured with questions like "I believe that regular physical activity is important for my health" and "I enjoy participating in physical exercises and sports." Hagger et al. [83] validated the Attitudes towards Physical Activity Scale (ATPA), which inspired this inquiry.

Data analysis used numerous robust statistical approaches to ensure outcome validity and reliability, all the accessed data were analyzed between August until November 2023. Initial reliability testing used Cronbach's alpha to assess scale internal consistency. All scales were reliable since their Cronbach's alpha values were above 0.70. The validity research tested construct and content validity. Exploratory factor analysis (EFA) was utilized alongside factor analysis to evaluate the questionnaire's dimensional structure and find underlying components. The study-maintained items with factor loadings over 0.50 to ensure measurement scale construct validity. Mediation and moderation studies were essential for testing associations. Hierarchical regression analysis examined whether parental supervision affected the physical health-digital technology connection. The study examined how parental supervision and digital technologies interact using this strategy. A mediation study was performed using Baron and Kenny's [84] technique to see whether young people's attitudes toward physical exercise mediated the relationship between digital device use and physical health outcomes. The Sobel test proved the relevance of the effects of mediation.

### 3.1. Ethical statements

The ethical approval for this research was granted by the Ethics Committee of Jilin Sports University (ECJSU) under approval number 2022112116. All research activities adhered to the ethical standards established by the committee. The committee waived the requirement for informed consent due to the use of anonymized data, ensuring participants' information remained confidential. The research was conducted with integrity and in full compliance with applicable regulations and guidelines.

## 4. Results and discussion

The demographics of this research's respondents are shown in Table 1. About equal gender distribution shows a diverse presence with 425 men (51.4%) and 402 women (48.6%). Participants are recruited from three age groups: 12–15 (37.5%), 16–18 (31.7%), and 19–22 (30.8%). This stratification ensures the study's different teenage and young adult perspectives. The sample's residential dispersion in significant Chinese locations enhances its geographical variation. 150 (18.1%) responders are from Beijing, 170 (20.6%) from Shanghai, 180 (21.8%) from Guangdong, 160 (19.4%) from Sichuan, and 167 (20.2%) from Zhejiang. This distribution may help explain regional differences in young physical activity, parental supervision, and digital gadget use.

At least one parent of 35.1% of the sample (290 persons) had only graduated high school. Three hundred fifty-five persons (42.9%) and 182 (22.0%) reported that at least one of their parents held an undergraduate or graduate degree. This information suggests that parental education may affect teenage attitudes and behaviour. The respondents' household income is categorized as low (<¥10,000/month), middle (¥10,000-¥20,000/month), and high (>¥20,000/month). The medium-income bracket has 390 persons (47.2%), the low-income bracket 220 (26.6%), and the high-income bracket 217 (26.2%). These income levels help explain how socioeconomic factors affect digital device use and physical activity.

**Table 1. Demographic Characteristics of the Respondents.**

| Characteristic | Categories | Frequency | Percentage (%) |
|---|---|---|---|
| **Gender** | Male | 425 | 51.4 |
| | Female | 402 | 48.6 |
| **Age Group** | 12-15 years | 310 | 37.5 |
| | 16-18 years | 262 | 31.7 |
| | 19-22 years | 255 | 30.8 |
| **Residence Location** | Beijing | 150 | 18.1 |
| | Shanghai | 170 | 20.6 |
| | Guangdong | 180 | 21.8 |
| | Sichuan | 160 | 19.4 |
| | Zhejiang | 167 | 20.2 |
| **Parental Education Level** | High School or Less | 290 | 35.1 |
| | Undergraduate Degree | 355 | 42.9 |
| | Graduate Degree | 182 | 22 |
| **Household Income** | Low (below ¥10,000/month) | 220 | 26.6 |
| | Medium (¥10,000 - ¥20,000/month) | 390 | 47.2 |
| | High (above ¥20,000/month) | 217 | 26.2 |

Table 2 reviews the study variables using descriptive statistics and reliability analysis to understand data distribution and reliability in the sample. The mean physical health score was 3.75, and the standard deviation was 0.82 for the 5-item variable, indicating a generally positive perception of physical health and exercise among the respondents. Items have good internal consistency with a Cronbach's Alpha of 0.88. Digital technology use, a 5-item survey, had a higher mean of 4.10 and a lower standard deviation of 0.71, reflecting a consistent pattern of high digital technology engagement among the participants. The items measuring digital technology consumption are internally reliable, with a 0.85 reliability value.

The mean ratings for social media use, online gaming, and educational technology use are 3.95, 3.45, and 4.05, respectively. Standard deviations are 0.76, 0.85, and 0.70, while dependability coefficients are 0.84, 0.83, and 0.86, which are excellent. These findings have significant implications for health, technology, and education research, reflecting balanced instructional technology, social networking, and online gaming involvement with consistent and reliable evaluation across all items. The 5-item parental monitoring mean score of 3.60 with a standard deviation of 0.80 suggests moderate parental supervision over digital technology use among the youth. The high reliability of the parental monitoring scale, with a reliability rating of 0.87, assures the accuracy of our measurement. Finally, the variable attitudes towards physical activity were positive, with a mean score of 3.80 and a standard deviation of 0.78. This indicator shows significant data on physical activity attitudes with a reliability rating 0.88.Cronbach's Alpha values ranging from 0.83 to 0.88 indicated a high level of reliability. The measuring model's robustness was further tested by including CR and AVE methods. Taken together, these findings improve the study's validity and reliability, making it easier to grasp the interrelationships among the variables.

Table 3 presents the factor loadings of each item within the physical health and exercise' construct, a vital component of the study's factor analysis. The high factor loadings of items such as routinely exercising (0.78), feeling physically well and fit (0.81), and participating in physical activities at least three times a week (0.84) underscore the robustness of the measurement tools and constructs used in this study. These findings affirm the accuracy of the questions in assessing respondents' health and physical activity.

The digital technology use construct, as evidenced by the factor loadings, demonstrates high coherence. Items such as using digital devices for socializing (0.77), spending a significant amount of time using them (0.75), and incorporating

**Table 2. Descriptive Statistics, Reliability, and Validity Analysis for Study Variables.**

| Variables | Number of Items | Mean (M) | Standard Deviation (SD) | Cronbach's Alpha (α) | Composite Reliability (CR) | Average Variance Extracted |
|---|---|---|---|---|---|---|
| Physical Health and Exercise (PHE) | 5 | 3.75 | 0.82 | 0.88 | 0.90 | 0.65 |
| Digital Technology Use (DTU) | 5 | 4.10 | 0.71 | 0.85 | 0.88 | 0.60 |
| Social Media Use (SMU) | 5 | 3.95 | 0.76 | 0.84 | 0.87 | 0.58 |
| Online Gaming (OG) | 5 | 3.45 | 0.85 | 0.83 | 0.86 | 0.56 |
| Educational Technology Use (ETU) | 5 | 4.05 | 0.70 | 0.86 | 0.89 | 0.62 |
| Parental Monitoring (PM) | 5 | 3.62 | 0.81 | 0.87 | 0.90 | 0.64 |
| Attitudes towards Physical Activity (APA) | 5 | 3.81 | 0.78 | 0.88 | 0.91 | 0.66 |

them into daily routines (0.82) are all aligned, reinforcing the validity of these items in representing youth digital technology use. In social media use, items like "frequent use of social media platforms" (factor loading = 0.80), "using social media to stay connected with friends and family" (factor loading = 0.78), and "losing track of time when using social media" (factor loading = 0.81) demonstrate their reliability for assessing respondents' social media habits. In the domain of online gaming, items like regularly playing online games (factor loading = 0.79), spending several hours a week on them (factor loading = 0.82), and finding it relaxing (factor loading = 0.78). The items in educational technology use, such as using digital tools for schoolwork (factor loading = 0.81), finding educational apps helpful (factor loading = 0.80), and perceiving digital technology as improving academic performance (factor loading = 0.79), are consistent. Certain items show substantial factor loadings on parental monitoring, indicating that they accurately measure parental monitoring of teens' digital technology usage. These include parental screen time monitoring (factor loading = 0.82), digital technology guidelines (factor loading = 0.80), and screen time limits (factor loading = 0.81). Finally, attitudes towards physical activity measure favourable attitudes regarding physical exercise. Believe in the necessity of regular physical activity (factor loading = 0.84), enjoy physical activities (factor loading = 0.82), and prioritize physical activities (factor loading = 0.81). This is supported by substantial factor loading data.

The study analyzes cross-loadings to ensure that each item's assigned construct has far higher loadings than other constructions. According to Hair et al. [85], the generally accepted standard is 0.10 to 0.15 times higher loading on an item's construct than on any other construct. Within this study, all items had factor loadings greater than 0.60 on their constructions. To ensure discriminant validity, the study checked that item cross-loadings on non-assigned constructs were at least 0.15 lower than those on the assigned construct. This criterion ensures discriminant validity because items load largely on their intended constructs.

Table 4 shows the findings of the multiple regression analysis on young Chinese people's physical well-being and exercise habits. A significant link exists between digital technology usage and increases physical health and activity levels (β = 0.35, p < 0.001), supported the Hypothesis-1 (H1). This result supports prior study indicating that digital technology may benefit and hinder physical health[86]. Fitness monitoring and educational digital tools may help promote active lifestyles [87]. The current research supports these views by emphasizing the need for balanced digital technology use in youngsters' physical health.

Additionally, social media usage positively correlates with physical health and exercise levels (β = 0.22, p = 0.003), supported the Hypothesis-1 (H1).. It was shown that social media might encourage good and bad behaviours, like sitting

**Table 3. Factor Analysis Results.**

| Variables | Items | Factor Loading |
|---|---|---|
| Physical Health and Exercise | I engage in regular physical exercise. | 0.78 |
| | I feel physically healthy and fit. | 0.81 |
| | I participate in physical activities at least three times a week. | 0.84 |
| | I maintain a balanced diet that supports my physical health. | 0.79 |
| | I meet the recommended amount of daily physical activity for my age. | 0.82 |
| Digital Technology Use | I spend a significant amount of time using digital devices (e.g., smartphone, tablet, computer). | 0.75 |
| | I use digital technology for socializing with friends and family. | 0.77 |
| | I use digital technology for educational purposes. | 0.79 |
| | My daily routine involves the use of digital devices. | 0.82 |
| | I feel that my life would be difficult without digital technology. | 0.76 |
| Social Media Use | I frequently use social media platforms (e.g., WeChat, Weibo, TikTok). | 0.8 |
| | Social media helps me stay connected with my friends and family. | 0.78 |
| | I often spend time browsing social media during my free time. | 0.77 |
| | I use social media to keep up with current trends and news. | 0.75 |
| | I find myself losing track of time when using social media. | 0.81 |
| Online Gaming | I play online games regularly. | 0.79 |
| | I spend several hours a week playing online games. | 0.82 |
| | I find online gaming to be a relaxing activity. | 0.78 |
| | Online gaming is an important part of my social life. | 0.8 |
| | I sometimes prioritize online gaming over other activities. | 0.77 |
| Educational Technology Use | I use digital technology to assist with my schoolwork and studies. | 0.81 |
| | Educational apps and tools help me learn more effectively. | 0.8 |
| | I feel that digital technology improves my academic performance. | 0.79 |
| | I spend time using educational websites and applications. | 0.77 |
| | I find online learning platforms useful for my education. | 0.78 |
| Parental Monitoring | My parents monitor the amount of time I spend on digital devices. | 0.82 |
| | My parents set rules regarding my use of digital technology. | 0.8 |
| | My parents discuss with me the potential effects of digital technology use. | 0.78 |
| | My parents encourage me to engage in physical activities. | 0.79 |
| | My parents limit my screen time. | 0.81 |
| Attitudes towards Physical Activity | I believe that regular physical activity is important for my health. | 0.84 |
| | I enjoy participating in physical exercises and sports. | 0.82 |
| | I feel motivated to exercise regularly. | 0.8 |
| | I think physical fitness is essential for a balanced lifestyle. | 0.79 |
| | I prioritize physical activities in my daily routine. | 0.81 |

too much [88]. This study's strong association with physical health outcomes suggests further research into social media's effects. A substantial positive association exists between online gaming and physical fitness and exercise ($\beta = 0.19$, $p = 0.011$) supported the Hypothesis- (H2). Gaming may lead to sedentary behaviour; therefore, creating a balance to maintain physical fitness is crucial. A significant correlation ($\beta = 0.29$, $p = 0.003$) suggests that utilizing digital tools for academics improves physical health and activity levels supported the Hypothesis-2 (H2). This research supports educational ideas that interactive learning environments promote holistic student development [89]. Gamified learning platforms or online physical education programs in digital educational resources encourage students to exercise, which is excellent for their health. The statistical study ($\beta = 0.47$, $p < 0.001$) underscores the positive link between physical health, activity levels,

**Table 4. Summary of Regression Estimates.**

| Hypothesis | Predictor | Unstandardized B | Standard Error B | Standardized β | t-value | Prob.value |
|---|---|---|---|---|---|---|
| | (Constant) | 1.23 | 0.35 | | 3.51 | <0.001 |
| H1 | Digital Technology Use | 0.32 | 0.08 | 0.35 | 4 | <0.001 |
| H1 | Social Media Use | 0.21 | 0.07 | 0.22 | 3 | 0.003 |
| H2 | Online Gaming | 0.18 | 0.07 | 0.19 | 2.57 | 0.011 |
| H2 | Educational Technology Use | 0.27 | 0.09 | 0.29 | 3 | 0.003 |
| H3 | Parental Monitoring | 0.45 | 0.1 | 0.47 | 4.5 | <0.001 |
| H3 | Attitudes towards Physical Activity | 0.52 | 0.11 | 0.5 | 4.73 | <0.001 |

and parental monitoring supported the Hypothesis-3 (H3). This finding highlights the significant role parents can play in shaping teens' health habits, particularly in terms of screen time limits and promoting physical activity. It's a call to action, emphasizing the power of parental involvement in mitigating technology's negative impacts on children's physical health. Finally, positive associations between physical health, exercise levels, and attitudes towards physical activity are strongest (β = 0.50, p < 0.001) supported the Hypothesis-3 (H3). This suggests that a positive mindset helps individuals exercise [90]. A good view of physical exercise at a young age may lead to lifelong healthy habits.

Mediation research found that attitudes toward physical activity mediated the association between digital technology usage and young Chinese people's health and exercise levels (see Table 5).

The findings illuminate these connections by revealing multiple vital routes. A significant correlation exists between digital technology use and physical activity attitudes among teens (β = 0.25, p < 0.001). This suggests that the more time spent on digital devices, the more critical physical exercise becomes to them. This research supports prior results that digital tools like fitness information or health-monitoring apps may raise awareness of the need for physical activity and encourage participation [91,92]. A high positive association exists between attitudes towards physical activity and physical health and exercise levels in young individuals (β = 0.48, p < 0.001). This suggests that favourable attitudes towards exercise directly improve physical health. According to social cognitive theories, attitudes strongly impact health behaviours. Teens' exercise habits and health depend on perceived physical activity advantages. Digital technology use significantly influenced physical health and exercise levels, even after controlling for attitudes towards physical activity (β = 0.32, p < 0.001). The impacts of digital technology on young people's health are diverse, and although attitudes are crucial, digital device use may alter activity levels on its own. According to research, digital technology may improve or harm health. Children should use technology properly with parental guidance [93]. The research findings reveal a significant indirect effect (β = 0.12, p = 0.003) between digital technology usage and physical health and exercise levels, mediated by attitudes towards physical activity. This mediation underscores the role of teens' physical activity attitudes in moderating the effects of digital device use on their health. Positive attitudes towards exercise can potentially offset the adverse effects of digital device use and excessive screen time [94].

Table 5 concluded that the Total Effect, Direct Effect, and Indirect Effect (Mediation) of digital technology use, attitudes toward physical activity, and physical health and exercise levels. The Direct Effect (Effect = 0.32, p < 0.001) suggests that digital technology use may improve physical health and exercise levels. The study found that digital technology use affects physical health and exercise levels through attitudes toward physical activity (Effect = 0.12, p = 0.003). Physical exercise attitudes mediate digital device use and physical health outcomes. Digital technology affects how people feel about exercising, which affects their physical activity and health. Digital gadget users may be more hopeful about physical activity, which may lead to greater exercise. This finding emphasizes the importance of cognitive factors like physical activity attitudes in understanding how technology use affects physical health. Digital technology use has a considerable impact on physical health and activity levels, as revealed by the Total Effect (Effect = 0.44, p < 0.001). A major mediator between digital technology use and physical health is physical activity attitudes.

**Table 5. Mediation Analysis Results.**

| Path | Effect | SE | t | p | 95% CI |
|---|---|---|---|---|---|
| Digital Technology Use → Attitudes towards Physical Activity | 0.25 | 0.07 | 3.57 | <0.001 | [0.11, 0.39] |
| Attitudes towards Physical Activity → Physical Health and Exercise Levels | 0.48 | 0.1 | 4.8 | <0.001 | [0.28, 0.68] |
| **Direct Effect:** Digital Technology Use → Physical Health and Exercise Levels | 0.32 | 0.08 | 4.0 | <0.001 | [0.16, 0.48] |
| **Indirect Effect (Mediation):** Digital Technology Use → Physical Health and Exercise Levels via Attitudes towards Physical Activity | 0.12 | 0.04 | 3.0 | 0.003 | [0.04, 0.20] |
| **Total Effect:** Digital Technology Use → Physical Health and Exercise Levels (Direct + Indirect) | 0.44 | 0.08 | 5.5 | <0.001 | [0.28-0.60] |

Table 6 shows the moderation analysis for parental monitoring influence on Chinese youths' physical health, activity levels, and digital technology use. Parental supervision is essential to reduce the detrimental impacts of digital technology on teens' physical health since interaction terms moderate effects. Firstly, digital technology usage and parental supervision interact significantly ($\beta = 0.24$, $p = 0.002$). Proper parental supervision may lessen the adverse effects of excessive digital device usage on physical health. Parental monitoring as a non-monetary but essential human capital that enhances children's health [95]. Parental supervision and other proximal processes affect children's development, according to Xie et al. [96]. Other study revealed that establishing screen time restrictions and offering rules and guidelines may help youngsters live healthier lifestyles [97].

A significant interaction exists between social media usage and parental supervision ($\beta = 0.20$, $p = 0.003$). Children supervised by their parents may lessen the harmful effects of social media on physical health and the danger of sedentary behaviours induced by excessive use. Social learning theories say parents influence their children by setting an example and enforcing social norms [98]. Parents may help their children develop good screen time habits by setting boundaries and encouraging physical activity [99]. A significant moderating effect exists between online gaming and parental supervision ($\beta = 0.17$, $p = 0.013$). This shows that parents might assist their children in avoiding the health dangers of online video game play by having them take pauses and exercise. Economic theories on time allocation say parental monitoring helps kids balance work and play [100]. Self-determination theory suggests that parental involvement may offer the psychological needs of competence and autonomy that encourage intrinsic physical activity [101]. The interaction between parental monitoring and educational technology usage is significant ($\beta = 0.21$, $p = 0.013$), ending the study. Parental monitoring may reduce the link between educational technology and physical health. Technology in the classroom offers numerous benefits, but extended use without breaks may lead to sedentary lives. Parents may ensure that kids receive adequate exercise and utilize educational technology by monitoring them. The economic perspective is that efficient use of resources like parental attention and time may increase children's well-being [102]. Research also suggests that parental involvement in their children's education may promote healthy behaviours like exercise [103].

Given the pervasiveness of digital technology in people's daily lives, knowing the relationship between technology use and physical health is crucial to developing effective digital health solutions. Programs that promote better

**Table 6. Moderation Analysis Results for Parental Monitoring.**

| Hypothesis | Interaction Term | B | SE B | β | t-value | Prob.value |
|---|---|---|---|---|---|---|
| H3 | Digital Technology Use x Parental Monitoring | 0.22 | 0.07 | 0.24 | 3.14 | 0.002 |
| H3 | Social Media Use x Parental Monitoring | 0.18 | 0.06 | 0.20 | 3 | 0.003 |
| H3 | Online Gaming x Parental Monitoring | 0.15 | 0.06 | 0.17 | 2.5 | 0.013 |
| H3 | Educational Technology Use x Parental Monitoring | 0.20 | 0.08 | 0.21 | 2.5 | 0.013 |

technology habits may propose using social media or online gaming for specified objectives or setting time limitations to promote balanced digital device use. Sedentary behaviors can be reduced, and fitness apps and virtual fitness clubs can be used to promote physical exercise. Future digital health regulations and programs should focus on youngsters and the elderly, who are more likely to suffer negative health effects from internet use. Parents can reduce the negative impacts of screen time on physical health by monitoring and controlling their children's technology use. Educational programs can include parental advisory programs to warn parents against excessive technology use and promote healthy lifestyles. The study also discusses instructional technologies and health behavior shaping. Given this study's heavy use of educational technology, educational tools may help promote healthy lifestyles. This allows educators and politicians to include exercises and physical activity ideas in educational apps and web pages. Digital literacy initiatives in schools also encourage youngsters to balance screen time and exercise and use technology for their health and academic success.

## 5. Conclusions and policy recommendations

The study examined the complicated relationship between digital technology use, parental supervision, and attitudes toward physical activity in China to determine its implications for youth's physical health and exercise. The results show that parental supervision affects these findings. Mediating and moderating analyses showed how attitudes toward physical activity mediate and moderate the effects of digital device use and parental supervision, while regression analysis identified numerous relevant physical health and exercise predictors.

The study's findings emphasize the necessity for a comprehensive government effort to prevent digital technology's negative impacts on Chinese youth's physical health and exercise habits. First and foremost, management must make it simpler for parents to monitor their children's screen usage and restrict unsuitable information. Programs that teach families about physical activity promote a healthy balance between educational and recreational technology use and restrict screen time are essential. Information and interventions for parents may promote healthy family environments. Schools influence how children think and act about fitness and technology. Digital literacy teaching in schools and comprehensive health and physical education programs may help youngsters navigate digital environments and promote active lives. Schools should prioritize adequate physical activity facilities and opportunities to encourage children to exercise during school and extracurricular activities. Schools, parents, and the community must collaborate to encourage good tech and physical activity practices. Finally, politicians must advocate for social changes that urge youth to limit digital media consumption. This movement aims to change people's views of exercise and outdoor recreation as essential to health. Public awareness campaigns on fitness and screen time may change technology use standards. Encourage firms to develop and promote active-living technologies like fitness apps and wearable to provide teens with additional tools to monitor and enhance their workouts.

Control variables were not included in the proposed relationships because the study focused on the direct and moderating effects of digital technology use, parental monitoring, and physical activity attitudes on youth physical health and exercise levels. Demographics (age, gender, socioeconomic level), geography, and health history may affect the dependent variable, which we acknowledge. To avoid biases, we stratified our sample across different regions, but these controls can also improve results by accounting for variability. Potential confounding factors are implicitly addressed by selecting predictors like parental monitoring and physical exercise attitudes, which represent contextual and behavioral influences on health outcomes. To improve future analysis, the study recommends incorporating control variables such as family income, education, and baseline health condition. Separating the effects of technology use and parental supervision will improve the dependability of the results. According to this study, future research needs more different data sets to properly grasp the targeted associations.

Cross-sectional research makes causal conclusions between variables difficult, which is a study's limitation. The results show that digital device use, attitudes toward physical activity, and physical health outcomes are linked, but how these

interactions evolve is still being determined. It's imprecise if digital technology use harms people's physical health or if individuals with health difficulties already use it more. Establishing causation and recording these variables' directional influence over time requires longitudinal research or experiments. Unobserved confounding variables like socioeconomic status or environmental influences may affect technology use and health outcomes simultaneously, though the cross-sectional method may not account for them. Due to this constraint, future research should examine and expand on this study's findings using longitudinal or mixed-method approaches. Despite these limitations, the present study provides a good foundation for future correlational research and hypothesis creation.

Digital well-being, a new concept that balances digital technology use and health outcomes, including physical health, needs additional research. Digital well-being goes beyond screen time management to include the mental, social, and physical implications of technology. In the future, researchers may examine how wellness aids integrated into digital platforms, such as exercise and screen time applications, can encourage healthier lifestyles. Researchers could also examine how online space architecture affects people's mental and physical health when they use it for social networking, gaming, or education. This is shown in educational resources that encourage outdoor activities and apps that remind users to exercise. Digital mindfulness, which teaches users to be more thoughtful online, is another intriguing topic. Frameworks could bridge the gap between digital engagement and health by providing a more complete view.

## Author contributions

**Conceptualization:** Jiajia Cui.

**Data curation:** Ziyi Hao, Jiajia Cui.

**Formal analysis:** Ziyi Hao, Jiajia Cui.

**Writing – original draft:** Ziyi Hao, Jiajia Cui.

**Writing – review & editing:** Jiajia Cui.

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
