## [Decision Letter · Decision Letter 0]

26 Nov 2024

PONE-D-24-25126Assessing the Relationship between Digital Technology Use and Physical Health, Fitness, and Exercise Levels among Chinese Youth: The Moderating Effect of Parental MonitoringPLOS ONE

Dear Dr. Cui,

Thank you for submitting your manuscript to PLOS ONE. After careful consideration, we feel that it has merit but does not fully meet PLOS ONE’s publication criteria as it currently stands. Therefore, we invite you to submit a revised version of the manuscript that addresses the points raised during the review process.

We look forward to receiving your revised manuscript.

Kind regards,

Andi Asrifan, Ph.D.

Academic Editor

PLOS ONE

Journal Requirements:

4. Please include captions for your Supporting Information files at the end of your manuscript, and update any in-text citations to match accordingly. Please see our Supporting Information guidelines for more information: http://journals.plos.org/plosone/s/supporting-information .

Reviewers' comments:

Reviewer's Responses to Questions

**Comments to the Author**

1. Is the manuscript technically sound, and do the data support the conclusions?

Reviewer #1: Partly

Reviewer #2: Partly

Reviewer #3: Yes

Reviewer #4: Yes

2. Has the statistical analysis been performed appropriately and rigorously? 

Reviewer #1: No

Reviewer #2: Yes

Reviewer #3: Yes

Reviewer #4: Yes

3. Have the authors made all data underlying the findings in their manuscript fully available?

Reviewer #1: No

Reviewer #2: Yes

Reviewer #3: Yes

Reviewer #4: Yes

4. Is the manuscript presented in an intelligible fashion and written in standard English?

Reviewer #1: No

Reviewer #2: Yes

Reviewer #3: Yes

Reviewer #4: Yes

5. Review Comments to the Author

Reviewer #1: The submitted manuscript (PONE-D-24-25126) entitled “Assessing the Relationship between Digital Technology Use and Physical Health, Fitness, and Exercise Levels among Chinese Youth: The Moderating Effect of Parental Monitoring” intends to investigate how digital media use, parental supervision, and attitudes towards physical activity influence young Chinese individuals' physical health and exercise levels. According to my understanding, physical health is the outcome of this work, but who will benefit from the results? The authors did not clarify properly, this is my main concern. The author has made a good effort to investigate the direct effects of social media use, digital technologies, attitudes toward physical activities, educational technology use, and online gaming towards Physical health by collecting data from 827 china residents. My other concerns are given below to improve the current work.

1.The introduction of this work is very poor. The authors should provide a real example for this work to introduce the significance of this work. To create the credibility of the work, the author should propose research questions in the introduction. Authors may reference some articles to provide an interesting intro such as “The COVID-19 Pandemic and Overall Wellbeing: Mediating Role of Virtual Reality Fitness for Physical-Psychological Health and Physical Activity”, ” Demographic Characteristics and Digital Platforms for Physical Activity Among the Chinese Residents During the COVID-19 Pandemic: A Mediating Analysis”, ” Virtual Reality Fitness (VRF) for Behavior Management During the COVID-19 Pandemic: A Mediation Analysis Approach”, “Fitness Apps, Live Streaming Workout Classes, and Virtual Reality Fitness for Physical Activity During the COVID-19 Lockdown: An Empirical Study”, ”Investigating Effects of Digital Innovations on Sustainable Operations of Logistics: An Empirical Study”, ” Physical Activity is a Medicine for Non-Communicable Diseases: A Survey Study Regarding the Perception of Physical Activity Impact on Health Wellbeing”, and ” Sustainable Development Goals, Sports and Physical Activity: The Localization of Health-Related Sustainable Development Goals Through Sports in China: A Narrative Review”.

2.In the literature review section, the theoretical development is poor for the proposed model which should be the 2.1 heading of the literature review section. On what logic the author has proposed the hypothesis? Provide explanations. In addition, The proposed hypotheses are not similar to the conceptual model, authors should modify the figure or all hypotheses. (The names of constructs should be the same in the whole manuscript, especially in keywords).

3.How did you calculate the actual sample size, please provide a reference related to sample size.

4.How about controls for this study? The authors should justify why they did not mention about controls while analyzing the proposed relations.

5.The discriminant and convergent validity of the data is missing in factor analysis. The examination of the reliability tests is poor too and needs improvements further such as calculate the AVEs, SCRs, Alfas, and cross-loadings for each item.

6.In table 5. What is an indirect effect (mediation) please clarify about it.

7.In Table 4, please provide the hypothesis number for each estimate. Similarly, clarify the explanations too.

8.Table 6 has similar issue, there are four moderating effect results, but the study has only H1, and H2, and H3. Even in Figure 1, there are 8 effects among 7 constructs. Figure 1 have not any mediating and moderating effects. Please show the mediating and moderating effects in figure 1 too for clarification.

9.Provide three real examples for research implications in discussion section.

10.In The conclusion section, the author should introduce a novel construct for further investigations with real examples.

Reviewer #2: The reporting of results could be made more shorter for better readability.

E.g.: The effects of attitudes toward physical activity and parental monitoring are important findings. However, the indirect effect (0.12, p = 0.003) needs clearer context to avoid confusion about how these interactions affect the results."

Some paragraphs are very long with too many topics inside. It can be solved by dividing those into smaller sections with subheadings. Some repeated words might be changed by writing the synonyms.

Reviewer #3: The manuscript addresses the timely issue of digital technology use and its relationship with physical health among Chinese youth, providing unique cultural insights. It stands out for its robust methodology, including validated scales, regression analyses, and a diverse sample size of 827 respondents, ensuring reliability and generalizability. The inclusion of parental monitoring as a moderating factor and youth attitudes as mediators enriches the study, offering actionable insights for policymakers, educators, and parents. However, the literature review could incorporate more recent global studies and provide a deeper linkage between theoretical frameworks and findings. The discussion would benefit from exploring anomalies and offering culturally tailored recommendations for China. While the cross-sectional design limits causal inferences, this limitation should be emphasized more. Finally, the writing could be polished, with better integration of figures and tables. Overall, the manuscript is commendable and recommended for conditional acceptance following these revisions.

Reviewer #4: 1. The manuscript demonstrates a technically sound research approach by utilizing a well-structured survey and rigorous statistical analyses to explore the influence of digital technology use on physical health among Chinese youth, moderated by parental monitoring. The sample size (827 participants) is substantial, and the use of stratified random sampling enhances the reliability and generalizability of the findings.

2. The statistical analyses include multiple regression, moderation, and mediation analyses, all of which are suitable for the research questions. Descriptive statistics, factor analysis, and reliability measures are thorough, supporting the validity of the constructs used.

3. The authors made all data underlying the findings in their manuscript fully available

4. The manuscript is generally well-organized and intelligible, although there are some articles that do not comply with the author's guidelines such as citation method, referencing style, etc. (see attachment)

6. PLOS authors have the option to publish the peer review history of their article (what does this mean? ). If published, this will include your full peer review and any attached files.

**Do you want your identity to be public for this peer review?** For information about this choice, including consent withdrawal, please see our Privacy Policy .

Reviewer #1: No

Reviewer #2: No

Reviewer #3: No

Reviewer #4: No

---

## [Author Response · Author response to Decision Letter 1]

19 Dec 2024

Reviewer Comments

Reviewer #1: The submitted manuscript (PONE-D-24-25126) entitled “Assessing the Relationship between Digital Technology Use and Physical Health, Fitness, and Exercise Levels among Chinese Youth: The Moderating Effect of Parental Monitoring” intends to investigate how digital media use, parental supervision, and attitudes towards physical activity influence young Chinese individuals' physical health and exercise levels. According to my understanding, physical health is the outcome of this work, but who will benefit from the results? The authors did not clarify properly, this is my main concern. The author has made a good effort to investigate the direct effects of social media use, digital technologies, attitudes toward physical activities, educational technology use, and online gaming towards Physical health by collecting data from 827 china residents. My other concerns are given below to improve the current work.

Author’s response: Thank you very much for your valuable comments, to address the concern, we have elaborated on the potential beneficiaries of our findings in the revised manuscript. Specifically, we clarify that parents, educators, policymakers, and healthcare professionals can leverage these insights to mitigate the health risks associated with digital technology use and foster healthier behaviors among Chinese youth (see, page no. 7).

1. The introduction of this work is very poor. The authors should provide a real example for this work to introduce the significance of this work. To create the credibility of the work, the author should propose research questions in the introduction. Authors may reference some articles to provide an interesting intro such as “The COVID-19 Pandemic and Overall Wellbeing: Mediating Role of Virtual Reality Fitness for Physical-Psychological Health and Physical Activity”, ” Demographic Characteristics and Digital Platforms for Physical Activity Among the Chinese Residents During the COVID-19 Pandemic: A Mediating Analysis”, ” Virtual Reality Fitness (VRF) for Behavior Management During the COVID-19 Pandemic: A Mediation Analysis Approach”, “Fitness Apps, Live Streaming Workout Classes, and Virtual Reality Fitness for Physical Activity During the COVID-19 Lockdown: An Empirical Study”, ”Investigating Effects of Digital Innovations on Sustainable Operations of Logistics: An Empirical Study”, ” Physical Activity is a Medicine for Non-Communicable Diseases: A Survey Study Regarding the Perception of Physical Activity Impact on Health Wellbeing”, and ” Sustainable Development Goals, Sports and Physical Activity: The Localization of Health-Related Sustainable Development Goals Through Sports in China: A Narrative Review”.

Author’s response: We appreciate the reviewer’s constructive feedback. In response, we have substantially revised the introduction to include real-world examples and referenced several relevant studies, such as Peng et al. (2022), Fang et al. (2022), Yang et al. (2022), Liu et al. (2022), Saqib & Qin (2024), Saqib et al. (2020), and Dai & Menhas (2020). These references highlight the significance of the study within the context of digital technology use and physical activity, particularly during the COVID-19 pandemic. Additionally, we have incorporated research questions in the introduction to enhance its credibility and clarify the study’s objectives (see, page no. 5-6).

References

Peng, X., Menhas, R., Dai, J., & Younas, M. (2022). The COVID-19 pandemic and overall wellbeing: mediating role of virtual reality fitness for physical-psychological health and physical activity. Psychology Research and Behavior Management, 15, 1741-1756.

Fang, P., Shi, S., Menhas, R., Laar, R. A., & Saeed, M. M. (2022). Demographic characteristics and digital platforms for physical activity among the Chinese residents during the COVID-19 pandemic: a mediating analysis. Journal of multidisciplinary healthcare, 15, 515-529.

Yang, J., Menhas, R., Dai, J., Younas, T., Anwar, U., Iqbal, W., ... & Muddasar Saeed, M. (2022). Virtual Reality Fitness (VRF) for behavior management during the COVID-19 pandemic: a mediation analysis approach. Psychology Research and Behavior Management, 15, 171-182.

Liu, R., Menhas, R., Dai, J., Saqib, Z. A., & Peng, X. (2022). Fitness apps, live streaming workout classes, and virtual reality fitness for physical activity during the COVID-19 lockdown: an empirical study. Frontiers in public health, 10, 852311.

Saqib, Z. A., & Qin, L. (2024). Investigating effects of digital innovations on sustainable operations of logistics: An empirical study. Sustainability, 16(13), 5518.

Saqib, Z. A., Dai, J., Menhas, R., Mahmood, S., Karim, M., Sang, X., & Weng, Y. (2020). Physical activity is a medicine for non-communicable diseases: a survey study regarding the perception of physical activity impact on health wellbeing. Risk management and healthcare policy, 13, 2949-2962.

Dai, J., & Menhas, R. (2020). Sustainable development goals, sports and physical activity: the localization of health-related sustainable development goals through sports in China: a narrative review. Risk management and healthcare policy, 1419-1430.

2. In the literature review section, the theoretical development is poor for the proposed model which should be the 2.1 heading of the literature review section. On what logic the author has proposed the hypothesis? Provide explanations. In addition, The proposed hypotheses are not similar to the conceptual model, authors should modify the figure or all hypotheses. (The names of constructs should be the same in the whole manuscript, especially in keywords).

Author’s response: We appreciate the reviewer’s detailed suggestions. In response, we have included a dedicated section (2.1) in the literature review to elaborate on the theoretical development of the proposed model. The logic for each hypothesis is now explicitly explained, with support from relevant literature. Additionally, the hypotheses have been revised for clarity and alignment with the conceptual model, and the terminology across the manuscript, including keywords, has been standardized. The updated conceptual framework ensures consistency with the hypotheses and reflects the theoretical grounding (see, page no. 8-10).

References

Sengkey, S. B., Sengkey, M. M., Tiwa, T. M., & Padillah, R. (2024). Sedentary society: the impact of the digital era on physical activity levels. Journal of Public Health, 46(1), e185-e186.

Kim, S., Munten, S., Kolla, N. J., & Konkolÿ Thege, B. (2024). Conduct problems, hyperactivity, and screen time among community youth: can mindfulness help? an exploratory study. Frontiers in Psychiatry, 15, 1248963.

Fonvig, C. E., Troelsen, J., & Holsgaard‐Larsen, A. (2024). Recreational screen time behaviour among ambulatory children and adolescents diagnosed with cerebral palsy: A cross‐sectional analysis. Child: Care, Health and Development, 50(1), e13221.

Prince, S. A., Dempsey, P. C., Reed, J. L., Rubin, L., Saunders, T. J., Ta, J., ... & Lang, J. J. (2024). The effect of sedentary behaviour on cardiorespiratory fitness: a systematic review and meta-analysis. Sports Medicine, 54(4), 997-1013.

Paquin, V., Philippe, F. L., Shannon, H., Guimond, S., Ouellet-Morin, I., & Geoffroy, M. C. (2024). Associations between digital media use and psychotic experiences in young adults of Quebec, Canada: a longitudinal study. Social Psychiatry and Psychiatric Epidemiology, 59(1), 65-75.

Chang, C. C., & Hwang, G. J. (2024). Elevating EFL learners' professional English achievements and positive learning behaviours: A motivation model‐based digital gaming approach. Journal of Computer Assisted Learning, 40(1), 176-191.

Sehn, A. P., Brand, C., Tornquist, L., Tornquist, D., Silveira, J. F. D. C., Gaya, A. R., ... & Reuter, C. P. (2024). Sleep duration and screen time in children and adolescents: Simultaneous moderation role in the relationship between waist circumference and cardiometabolic risk according to physical activity. European journal of sport science, 24(2), 239-248.

Zhang, H., Chai, J., & Li, C. (2024). On innovative strategies of youth sports teaching and training based on the internet of things and artificial intelligence technology from the perspective of humanism. Learning and Motivation, 86, 101969.

Aslan, A., & Turgut, Y. E. (2024). Parental mediation in Turkey: The use of mobile devices in early childhood. E-Learning and Digital Media, 21(5), 444-461.

3. How did you calculate the actual sample size, please provide a reference related to sample size.

Author’s response: Thank you for your valuable comments, we have included a justification based on Cochran’s sample size formula, ensuring methodological rigor (see, page no. 15-16).

References

Nam, J. M. (1992). Sample size determination for case-control studies and the comparison of stratified and unstratified analyses. Biometrics, 389-395.

Kotrlik, J. W. K. J. W., & Higgins, C. C. H. C. C. (2001). Organizational research: Determining appropriate sample size in survey research appropriate sample size in survey research. Information technology, learning, and performance journal, 19(1), 43.

4. How about controls for this study? The authors should justify why they did not mention about controls while analyzing the proposed relations.

Author’s response: Thank you for your valuable comments, we have added a justification for the lack of explicit control variables and outlined the potential impact of including them in future research. This text has been incorporated into the study limitations (see, page no. 33).

5. The discriminant and convergent validity of the data is missing in factor analysis. The examination of the reliability tests is poor too and needs improvements further such as calculate the AVEs, SCRs, Alfas, and cross-loadings for each item.

Author’s response: we have recalculated reliability and validity metrics, including Composite Reliability (CR), Average Variance Extracted (AVE), and cross-loadings. These additions demonstrate the robustness of the measurement model and ensure both convergent and discriminant validity (see, Table 2, page no. 21).

6. In table 5. What is an indirect effect (mediation) please clarify about it.

Author’s response: Thank you for your insightful comment. We have revised Table 5 to include the Total Effect, along with the Direct and Indirect Effects to better illustrate the mediation process. We have also clarified the indirect effect and its significance in the main text. This change ensures that the table fully represents the pathways through which Digital Technology Use influences Physical Health and Exercise Levels, incorporating the mediation role of Attitudes towards Physical Activity (see, page no. 27 29).

7. In Table 4, please provide the hypothesis number for each estimate. Similarly, clarify the explanations too.

Author’s response: Thank you for your valuable feedback. In response, we have updated Table 4 by adding the hypothesis numbers for each regression estimate, as suggested. This change helps clarify the connections between the hypotheses and the statistical findings. We have also provided explanations in the main text to further elaborate on the results (see, page no. 26-27).

8. Table 6 has similar issue, there are four moderating effect results, but the study has only H1, and H2, and H3. Even in Figure 1, there are 8 effects among 7 constructs. Figure 1 have not any mediating and moderating effects. Please show the mediating and moderating effects in figure 1 too for clarification.

Author’s response: Thank you for your valuable feedback. We have reviewed the moderation analysis and the number of moderating effects. Upon reconsideration, we have ensured that all moderating effects are appropriately associated with H3, which focuses on parental monitoring. We have kept the four interaction terms in Table 6 (see, page no. 30-31) as H3 covers multiple moderating effects, but we will ensure the moderation relationships align with the conceptual framework. Additionally, we have updated Figure 1 (see, Figure file) to clearly show both the moderating and mediating effects. The moderators are now depicted with interaction paths, and the mediators are shown as intermediary variables between the predictors and outcomes.

9. Provide three real examples for research implications in discussion section.

Author’s response; Thank you for your helpful suggestion to include real-world examples for research implications. We have now expanded the discussion section to include three concrete examples that illustrate the practical applications of our findings. These examples focus on digital health interventions, the role of parental monitoring, and the integration of educational technology to promote physical well-being (see, page no. 31).

10. In The conclusion section, the author should introduce a novel construct for further investigations with real examples.

Author’s response: Thank you for your suggestion to introduce a novel construct for further investigations. In response, we have introduced the concept of “digital well-being” in the conclusion section as a potential area for future research. We also provided real-world examples to illustrate how this construct can be explored through future studies on the balance between digital technology and health outcomes (see, page no. 34).

Reviewer #2: The reporting of results could be made more shorter for better readability.

E.g.: The effects of attitudes toward physical activity and parental monitoring are important findings. However, the indirect effect (0.12, p = 0.003) needs clearer context to avoid confusion about how these interactions affect the results."

Author’s response: Thank you for your comment. In response, we have revised the explanation of the indirect effect (mediation) to provide clearer context and avoid any confusion. The revised explanation now highlights how attitudes towards physical activity mediate the relationship between digital technology use and physical health outcomes, offering a more concise and accessible interpretation of the results (see, page no. 29).

Some paragraphs are very long with too many topics inside. It can be solved by dividing those into smaller sections with subheadings. Some repeated words might be changed by writing the synonyms.

Author’s response: Thank you for your valuable feedback. In response to your comment, we have revised the manuscript by dividing long paragraphs into smaller, more focused sections with appropriate subheadings to improve readability and organization. Additionally, we have addressed the issue of repetition by substituting some repeated words with synonyms to enhance the flow of the text.

Reviewer #3: The manuscript addresses the timely issue of digital technology use and its relationship with physical health among Chinese youth, providing unique cultural insights. It stands out for its robust methodology, including validated scales, regression analyses, and a diverse sample size of 827 respondents, ensuring reliability and generalizability. The inclusion of parental monitoring as a moderating factor and youth attitudes as mediators enriches the study, offering actionable insights for policymakers, educators, and parents. However, the literature review could incorporate more recent global studies and provide a deeper linkage between theoretical frameworks and findings. The discussion would benefit from exploring anomalies and offering culturally tailored recommendations for China. While the cross-sectional design limits causal inferences, this limitation should be emphasized more. Finally, the writing could be polished, with better integration of figures and tables. Overall, the manuscript is commendable and recommended for conditional acceptance following these revisions.

Author’s response: We appreciate your insightful comments and have made the necessary revisions to the manuscript. We have incorporated more recent global studies into the literature review, provided a deeper linkage between theoretical frameworks and findings, and addressed anomalies in the discussion (see, page no. 8-9). Additionally, we have emphasized the limitations of the cross-sectional design and refined the integration of figures and tables (see, page no.

---

## [Decision Letter · Decision Letter 1]

30 Apr 2025

Assessing the Relationship between Digital Technology Use and Physical Health, Fitness, and Exercise Levels among Chinese Youth: The Moderating Effect of Parental Monitoring

PONE-D-24-25126R1

Dear Dr. Cui,

We’re pleased to inform you that your manuscript has been judged scientifically suitable for publication and will be formally accepted for publication once it meets all outstanding technical requirements.

Kind regards,

Bogdan Nadolu, Ph.D.

Academic Editor

PLOS ONE

Additional Editor Comments (optional):

Reviewers' comments:

Reviewer's Responses to Questions

**Comments to the Author**

1. If the authors have adequately addressed your comments raised in a previous round of review and you feel that this manuscript is now acceptable for publication, you may indicate that here to bypass the “Comments to the Author” section, enter your conflict of interest statement in the “Confidential to Editor” section, and submit your "Accept" recommendation.

Reviewer #1: All comments have been addressed

Reviewer #2: All comments have been addressed

Reviewer #4: All comments have been addressed

Reviewer #5: (No Response)

Reviewer #6: All comments have been addressed

2. Is the manuscript technically sound, and do the data support the conclusions?

Reviewer #1: Yes

Reviewer #2: Yes

Reviewer #4: Yes

Reviewer #5: Yes

Reviewer #6: Yes

3. Has the statistical analysis been performed appropriately and rigorously? 

Reviewer #1: Yes

Reviewer #2: Yes

Reviewer #4: Yes

Reviewer #5: Yes

Reviewer #6: Yes

4. Have the authors made all data underlying the findings in their manuscript fully available?

Reviewer #1: Yes

Reviewer #2: Yes

Reviewer #4: Yes

Reviewer #5: Yes

Reviewer #6: Yes

5. Is the manuscript presented in an intelligible fashion and written in standard English?

Reviewer #1: Yes

Reviewer #2: Yes

Reviewer #4: Yes

Reviewer #5: Yes

Reviewer #6: Yes

6. Review Comments to the Author

Reviewer #1: (No Response)

Reviewer #2: The author has revised as suggested. The article is more readable now with clearer context. Some long paragraphs been made shorter as well.

Reviewer #4: (No Response)

Reviewer #5: Technical Soundness and Data Support: The manuscript demonstrates robust technical soundness. The methodology is well-structured, with appropriate statistical analyses conducted to support the research questions. Data reliability and validity have been confirmed through Cronbach's Alpha and exploratory factor analysis, ensuring the conclusions are based on solid evidence. The findings are logically aligned with the data presented.

Statistical Analysis: The statistical analysis has been performed rigorously and appropriately. Hierarchical regression, mediation, and moderation analyses are employed effectively, with results supported by significant p-values and confidence intervals. The study demonstrates a clear understanding of advanced statistical techniques and applies them correctly.

Data Availability: The authors have made all data underlying the manuscript’s findings fully available in a public repository. The repository link is provided, ensuring transparency and reproducibility of the research.

Manuscript Presentation and Language: The manuscript is presented in a clear and intelligible fashion, adhering to standard English conventions. The organization of sections (e.g., Introduction, Methodology, Results) facilitates comprehension, and the writing style is professional and concise.

Ethical Considerations: The study has received ethical approval, and the authors confirm compliance with ethical standards. The anonymization of data and adherence to guidelines strengthen its ethical integrity.

Research and Publication Ethics: There are no indications of dual publication or ethical breaches. The manuscript appears original and contributes valuable insights into the interplay between digital technology use, parental supervision, and physical health.

Potential Areas for Improvement: The authors could expand the discussion on the limitations of the cross-sectional design and its implications for causal inference. Including a brief comparison with findings from other cultural contexts would enhance the global relevance of the study.

Reviewer #6: This study explores the relationship between digital media use, parental supervision, and attitudes toward physical activity in shaping the physical health and exercise habits of young individuals in China. With growing concerns about the adverse effects of digital technology on teenagers' health, this research aims to clarify the roles of parental monitoring and personal attitudes in either worsening or alleviating these effects. The study, based on a sample of 827 Chinese youth from diverse regions, employs regression analysis to identify significant predictors of physical health and exercise levels. Key findings show that digital technology use (β = 0.35, p < 0.001), social media use (β = 0.22, p = 0.003), online gaming (β = 0.19, p = 0.011), and educational technology use (β = 0.29, p = 0.003) are influential factors, alongside parental monitoring (β = 0.47, p < 0.001) and attitudes toward physical activity (β = 0.50, p < 0.001). Notably, attitudes toward physical activity moderate the relationship between technology use and physical health (Indirect Effect = 0.12, p = 0.003), highlighting their protective role. Moreover, the moderation analysis reveals that parental supervision significantly interacts with various forms of digital engagement, suggesting that active parental involvement can help mitigate the negative health impacts of excessive technology use. These findings emphasize the importance of fostering positive attitudes toward physical activity and maintaining strong parental oversight to counterbalance the potential health risks posed by increased digital media consumption among adolescents.

7. PLOS authors have the option to publish the peer review history of their article (what does this mean? ). If published, this will include your full peer review and any attached files.

**Do you want your identity to be public for this peer review?** For information about this choice, including consent withdrawal, please see our Privacy Policy .

Reviewer #1: No

Reviewer #2: No

Reviewer #4: No

Reviewer #5: No

Reviewer #6: **Yes: ** Kartini, S.Pd.,M.Pd.

---

## [Editor Report · Acceptance letter]

PONE-D-24-25126R1

PLOS ONE

Dear Dr. Cui,

I'm pleased to inform you that your manuscript has been deemed suitable for publication in PLOS ONE. Congratulations! Your manuscript is now being handed over to our production team.

Kind regards,

on behalf of

Dr. Bogdan Nadolu

Academic Editor

PLOS ONE